# Application of Standardized Regression Coefficient in Meta-Analysis

Pentti Nieminen

Medical Informatics and Data Analysis Research Group, University of Oulu, 90014 Oulu, Finland;
pentti.nieminen@oulu.fi

**Abstract:** The lack of consistent presentation of results in published studies on the association between a quantitative explanatory variable and a quantitative dependent variable has been a long-term issue in evaluating the reported findings. Studies are analyzed and reported in a variety of ways. The main purpose of this review is to illustrate the procedures in summarizing and synthesizing research results from multivariate models with a quantitative outcome variable. The review summarizes the application of the standardized regression coefficient as an effect size index in the context of meta-analysis and describe how it can be estimated and converted from data presented in original research articles. An example of synthesis is provided using research articles on the association between childhood body mass index and carotid intima-media thickness in adult life. Finally, the paper shares practical recommendations for meta-analysts wanting to use the standardized regression coefficient in pooling findings.

**Keywords:** standardized regression coefficient; statistics; meta-analysis; research synthesis; data presentation; carotid intima-media thickness; overweight; childhood

## 1. Introduction

Systematic reviews and meta-analyses are used to synthesize the available evidence for a given question in several scientific disciplines [1,2]. A review of the original articles and research synthesis extends our knowledge through the combination and comparison of the original studies. A major problem in analyzing, evaluating and summarizing the reported findings of studies on the association between a quantitative explanatory variable and a quantitative dependent variable is that the results are analyzed and reported in many ways [3–5]. When using a systematic literature review with a meta-analytical approach to learn from combined studies, we are dependent on the research methodology and reporting of the underlying studies. When the reviewed research articles contain inadequate statistical reporting of applied research methods and poor data presentation, the pooling of the findings will be even more difficult for the meta-analyst.

Among studies measuring the relationship between an explanatory factor and a response variable, some use correlation coefficients, some apply multivariable regression methods, and some studies compare mean values [5,6]. In addition, different measurement methods are used to assess the explanatory factors in the original studies. The quality of data presentation also varies. Detailed descriptive statistics of the variables under study are not given in all articles, and necessary measures of variation (standard errors) for coefficients of associations are not directly provided. Multivariable relationships present additional special challenges to meta-analysis because the statistics of interest depend on the other variables that are included in the multivariable analysis. Pooling these studies often requires data transformations and additional computations and estimations of effect sizes [1,7]. Thus, a coherent synthesis of studies analyzing the relation of an explanatory variable with a continuous outcome variable is challenging.

The measure used to represent the study findings in a meta-analysis is called an effect-size statistic. Several effect sizes have been proposed to synthesize results from multivariable regression models [8]. These include the unstandardized regression coefficient (*b*) and correlation coefficient (*r*). One effect-size approach is based on standardized regression coefficients. By definition, a standardized regression coefficient $\beta$ (also called a beta weight) represents the estimated number of standard deviations of change in the outcome variable for one standard deviation unit change in the explanatory or predictor variable, while controlling for other predictors. The synthesis of standardized regression coefficients has received attention over the last few decades because standardized regression coefficients are effect sizes commonly used in various domains [4,9–11]. Examples of applied disciplines include public-health and environmental research [12–14], psychology [15,16], and educational sciences [17,18].

As an example of using the standardized regression coefficient, I perform a meta-analysis to evaluate the association of childhood obesity and carotid intima-media thickness (cIMT) in adult life. Obesity induces multiple metabolic abnormalities that contribute to the pathogenesis of atherosclerosis and cardiovascular disease [19,20]. The carotid artery intima-media thickness is a marker of cardiovascular disease risk [21]. Thus, it is important to quantify the impact of childhood and adolescent body mass index (BMI) on common cIMT measurement in adulthood.

In this review, I first provide a description of the standardized regression coefficient ($\beta$) as an effect-size index. This is followed by a brief literature review of studies using these coefficients in different domains during the last ten years. An example is presented to illustrate the use of the meta-analysis technique for combining regression coefficients to synthesize findings from multivariable studies. The next chapter provides formulas to convert different statistics and effect sizes to standardized regression coefficients. After this, I discuss issues regarding the use of the standardized regression coefficient for combining effects. The main purposes of this paper are to point out the complexities and potential problems in a critical review of the association between a quantitative response variable and one primary quantitative explanatory variable, and to present a practical effect-size approach based on standardized regression coefficients.

## 2. Standardized Regression Coefficient as an Effect-Size Index in Meta-Analysis

Multivariable linear-regression models are used to analyze the associations between one quantitative dependent variable and several explanatory variables. The unstandardized regression coefficient (*b*) estimated from the linear-regression model is an easy-to-interpret statistic to describe how the explanatory variable affects the values of the outcome variable. These coefficients are usually provided with their standard errors (SEs) or confidence intervals (CIs) in articles reporting findings from regression models [22,23]. The unstandardized regression coefficient *b* describes the effect of changing the explanatory variable by one unit, and hence its size depends on the scale used to measure the explanatory variable. However, the main explanatory characteristic is often measured using different methods and metrics in the reviewed studies. Thus, the direct pooling of unstandardized regression coefficients is not meaningful across studies. To pool the effects of explanatory variables measured with different scales, we must express them in a comparable manner. In such a case, the standardized regression coefficient $\beta$ may offer an option to synthetize the findings [5,17]. The $\beta$ coefficient is the estimate resulting from an analysis carried out on variables that have been standardized so that their standard deviations (and variances) are equal to one [22,23]. Therefore, the standardized coefficient refers to how many standard deviations the response or outcome variable will change per a standard deviation increase in the explanatory or predictor variable. Thus, the standardized coefficient $\beta$ can be regarded as an attempt to make regression coefficients more comparable, and can be used as an effect-size estimate when the exposure levels in original studies are measured in different units of measurement.

The statistical significance of the standardized regression coefficient can be tested using the *t*-test of the null hypothesis $H_0$: $\beta = 0$, or in substantive terms, no systematic relationship between the predictor and outcome. A *p*-value higher than 0.05 supports the null hypothesis that there is no association. A confidence interval for the coefficient $\beta$ provides information about the range of the $\beta$. A positive (negative) β-value supports the hypothesis that a high exposure level increases (decreases) the response. When the confidence interval does not include 0, then the association between the explanatory variable and outcome variable is considered statistically significant, in accordance with the *p*-value of the *t*-test <0.05.

When considering effect sizes, a natural question to ask is what constitutes a large, medium, and small effect size. Cohen's [24] guidelines for the classification of effect sizes are widely cited in scientific reports. For a coefficient $\beta$, effect sizes between 0.10–0.29 are said to be only small, effect sizes between 0.30–0.49 are medium, and effect sizes of 0.50 or greater are large [24,25].

An essential feature of the quantitative meta-analysis is its ability to compare the magnitude of effects across studies, which requires the use of a single effect-size metric for measuring these effects. Using the standardized regression coefficient $\beta$ as the common effect-size measure involves extracting the findings of reviewed studies expressed as unstandardized regression coefficients, correlation coefficients or mean differences. These statistics are then re-expressed as standardized regression coefficients and their standard errors. This process includes several conversions, calculations, and approximations. The different approaches are summarized in Section 5.

In a meta-analysis, the findings (and effect sizes) are pooled from reviewed studies. However, every observed effect size is not equal with regard to the reliability of the information it carries [1]. Therefore, each effect-size value must be weighted by a term that represents its precision. An optimal approach is to use the inverse of the squared standard error of the effect-size value as a weight. Thus, larger studies, which have smaller standard errors, are given more weight than smaller studies, which have larger standard errors. The formula for computing the associated standard error must also be identified. To obtain the summary effect of all the reviewed studies, the weighted average effect size can be computed using the following formula:

$$M = \frac{\sum_{i=1}^{k} w_i \beta_i}{\sum_{i=1}^{k} w_i},$$

where $k$ = number studies, $\beta_i$ is the standard regression coefficient from study $I$, $SE(\beta_i)$ is the standard error of $\beta i$, and $w_i$ is the inverse of $(SE(\beta_i))^2$. The variance $(SE(\beta_i))^2$ can be calculated using the fixed-effects or random-effects model [1,26]. This version of the meta-analysis procedure is commonly referred to as the generic inverse-approach [27]. The approach is implemented in all standard software packages for meta-analysis.

Meta-analyses typically report the summary effect size $M$ with a measure of precision (SE or CI) and a *p*-value in a figure. This figure, the forest plot, displays the effect estimates and confidence intervals for individual studies as well as the summary effect. Figure 1 provides two examples of forest plots. Following Cohen's guidelines [24] and substantive empirical reviews [25,28], for the absolute (non-negative) value of the pooled effect size $|M|$, a value of 0.10–0.19 is a small effect size, a pooled value of 0.20–0.29 is classified as a medium effect size, and a pooled value of 0.30 or greater is a large effect size.

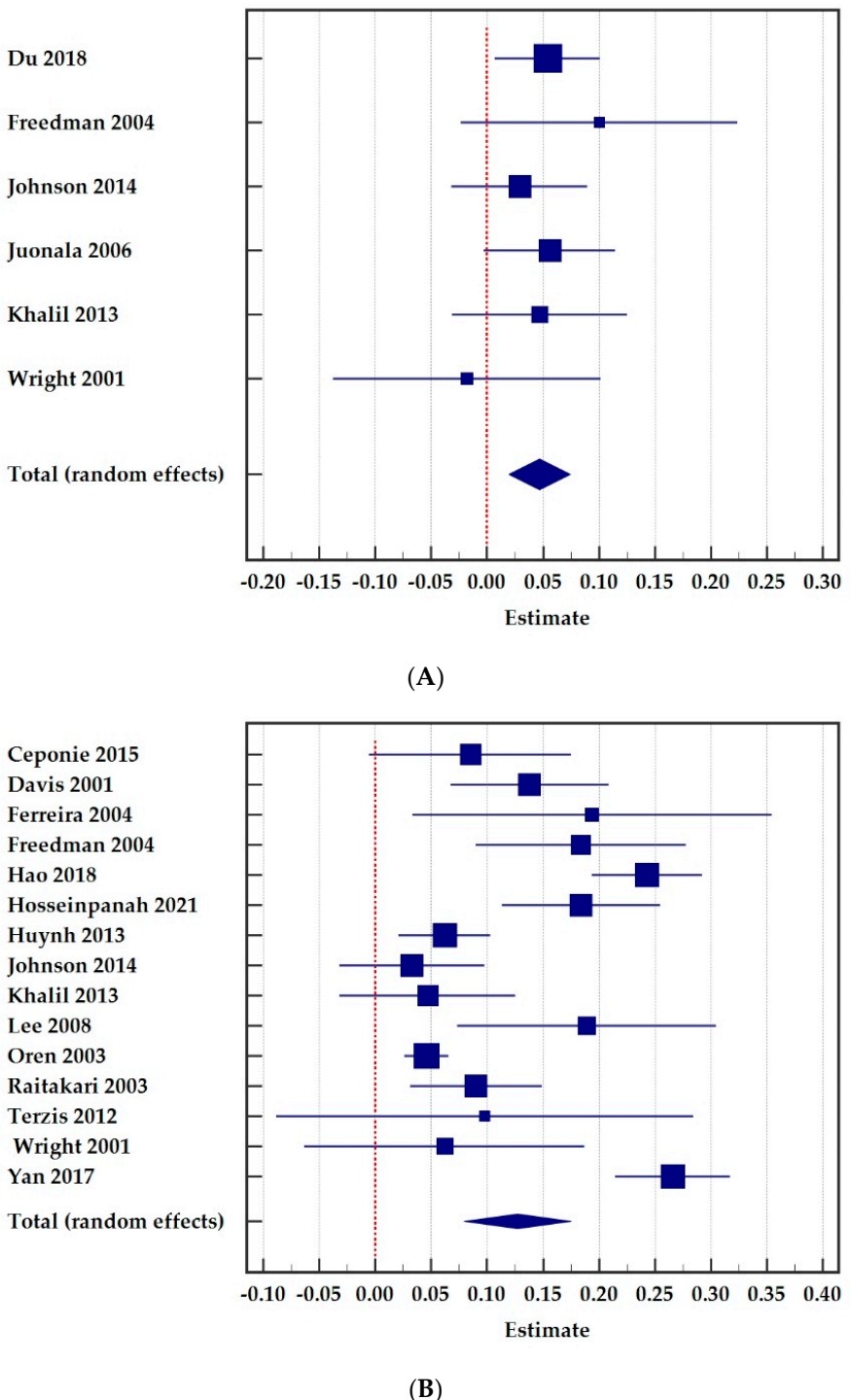

**Figure 1.** Forest plots for the association between childhood (**A**) and adolescent (**B**) and adult cIMT. Total number of individuals was 5796 in childhood and 11,859 in adolescent in the meta-analysis. (**A**) Childhood. (**B**) Adolescent.

## 3. Literature Review of Applications

In the following sub-chapters, I provide examples of meta-analytical studies where the use of the standardized regression coefficient served as a useful tool for synthesizing the results of numerous studies on a particular topic. Unfortunately, I also found meta-analyses where the coefficients *r*, *b* and *β* were confused [29–31].

### 3.1. Public Health

In environmental and public-health research, several outcomes and explanatory factors are often measured by different methods and units of measurement. Dzhambov and co-workers [13] studied whether green spaces and general greenery in the living environment of pregnant women were associated with the birth weight of their infants and what the direction of that effect was. They performed meta-analyses on eight published studies exploring the association of residential greenness and birth weight. The majority of the studies used multivariable linear regression to determine the effect of residential greenery on birthweight adjustments for personal covariates. In the original studies, different indicators were chosen as a proxy for residential greenness. Thus, the standardized regression coefficient offers one solution to pool the findings. The reported pooled $\beta$ was 0.001 (95% CI = $-0.001$ to 0.003), showing a non-significant association between greenness and birth weight. The authors noted that the findings were similar when the correlation coefficient was used as an effect-size index.

Keenan A. Ramsay and her co-authors [32] presented in their meta-analysis that higher physical activity (PA) and lower sedentary behavior (SB) are associated with greater skeletal muscle strength and muscle power in older adults. Articles were included in the meta-analyses if the associations between PA or SB measures and hand grip strength or the chair stand test were expressed as adjusted standardized regression coefficients ($\beta$) and their 95% CI or SE, or when these could be calculated. They identified considerable heterogeneity in the study design, the definitions of measures of outcome and explanatory variables, and the statistical analyses used to present the associations. This posed methodological challenges to comparing and synthesizing the results.

In healthy individuals and people with chronic pain, an inverse association between physical-activity level and pain has been reported (e.g., more activity and less pain). Jones et al. [33] examined the relation between aerobic capacity and pain in healthy individuals and people with fibromyalgia. They collated their new data with data from previous original studies in healthy individuals. To pool the findings identified by the literature search, standardized regression coefficients and their standard errors were calculated. This involved converting the results of analyses using the correlation, linear regression, or effect sizes of differences between groups and converting these to standardized $\beta$ coefficients with their standard errors. Then, 95% confidence intervals of the $\beta$s were calculated for presentation of the data on forest plots. Interestingly, the authors noted that a pooled effect size for these studies was not calculated, because they presented several effect sizes between various measures of pain and explanatory variables estimated from the same studies. Thus, the findings do not provide independent estimates of an effect. The presented forest plots (standardized $\beta$ coefficients with their 95% confidence intervals) of findings from studies illustrate clearly that the associations between physical fitness and pain are generally small and are highly variable within and across studies [33].

In 2020, Wang et al. [14] published a well-constructed quantitative summary of prenatal lead (Pb) exposure on birth weight. Because the quantitative variables from each reviewed article were reported using different metrics and different measures of association, they used standardized regression coefficients to allow a combination of findings from the reviewed studies. The pooling of findings was conducted separately for maternal blood and cord blood as measures of exposure variables. In addition, the analyses were restricted to unadjusted findings and to studies that adjusted for potential confounders. There was a significant negative association between prenatal Pb exposure and birth weight. In the unadjusted studies, birth-weight reduction was weakly associated with elevated lead levels in maternal blood (pooled $\beta = -0.094$, 95% CI = $-0.157$ to $-0.030$) and cord blood (pooled $\beta = -0.120$, 95% CI = $-0.239$ to $-0.001$). When restricted to the adjusted studies, these associations were weaker.

The study by Nicholas Burrows and his co-authors [34] reported meta-analyses of studies that examined correlations between pain from knee osteoarthritis and physical activity or fitness. The effect sizes from the evaluated original studies were converted

to standardized regression coefficients in order to be included on the forest plots and to estimate the pooled standardized coefficient. Data from their own new study were also included in the meta-analysis. From the 33 included studies, 13 provided data for the analysis of the associations between pain and physical activity, and 21 provided data for the associations between pain and fitness. The extracted physical-activity variables were either questionnaire-based measures of activity or objectively measured activity using pedometers or accelerometers. Separate meta-analyses were performed for muscle strength, muscle power, and aerobic capacity. Statistically significant pooled $\beta$s were found between objectively measured physical activity and pain severity. The more physically active individuals reported less pain at a baseline measurement, and across the seven-day period of physical-activity measurement.

McLaughlin et al. [35] reviewed studies related to the association between engagement with a physical-activity digital health intervention and physical-activity outcomes. A variety of different methods of association were used across the included studies. For the clearly reported meta-analysis, authors were required to transform several estimates into one consistent effect index. A standardized regression coefficient was chosen as the effect index. Many included studies reported more than one association. For meta-analyses, they used hierarchical selection criteria to select a single association from each study for inclusion in the pooled synthesis. When a study did not provide sufficient data required for meta-analysis (i.e., information to calculate an effect estimate and measure of variability of the effect estimate), the authors excluded this study from the meta-analysis. A meta-analysis of 11 included studies indicated a very small but statistically significant positive association between digital health engagement and physical activity (pooled $\beta = 0.08$, 95% CI = 0.01 to 0.14).

### 3.2. Psychology

Charlie Rioux and co-authors [16] published an interesting study where $\beta$s were used to represent the effect size of the interaction between temperament and family variables on substance use or externalizing behaviors while controlling for the other variables included in the tested model of the various studies. The authors searched for studies examining the interactions between temperament and the family environment on the outcome variables. Analyses of the interactions between two explanatory variables can be conducted using ANOVA techniques or with multiple regression models. The interpretation of the interactions is difficult because different patterns of interaction among temperament and family variables may have different implications. Due to issues with interaction terms and differences in measurements, the researchers were cautious and did not report pooled effect sizes. However, the reported individual effect sizes and their interpretation in the text still provide useful information about the possible interaction between the analyzed explanatory variables.

Kaitlin Woolley and Ayelet Fishbach [36] examined the relationship between immediate versus delayed rewards and persistence in long-term goals (e.g., healthy eating, exercising). The authors conducted five different intervention studies to examine the associations. In each study, they conducted a regression analysis to estimate the associations and reported $\beta s$. Finally, they pooled the $\beta$s using a meta-analytic approach to estimate an overall pattern across the five studies. In summary, whereas delayed rewards may motivate goal setting and the intentions to pursue long-term goals, a meta-analysis of their studies found that immediate rewards are more strongly associated with actual persistence in a long-term goal. The effect of immediate rewards on persistence, controlling for delayed rewards, was considered to be of medium size and statistically significant, (pooled $\beta = 0.35$, 95% CI = 0.28 to 0.42, $p < 0.001$).

Choi et al. [37] used a similar approach to Wooley and Fishbach [36] and combined the findings from five different studies using $\beta$ as the effect size. In each sub-study, they examined predictors of success in different achievement domains using regression models. By conducting meta-analyses, they explored the overall pattern across the studies. Their

findings indicate that self-control is predictive of success in achievement-related domains ($\beta$ = 0.27, 95% CI = 0.21 to 0.32), while emotional well-being is predictive of success in relationship-related domains ($\beta$ = 0.36, 95% CI = 0.29 to 0.43).

Two meta-analyses have examined the pain-related factors in individuals with chronic musculoskeletal pain [38,39]. In both studies, standardized regression coefficients and their 95% confidence intervals were calculated for the pooled results. Reviewed studies were excluded from these analyses if they did not provide sufficient information for computing the SE of the regression coefficient. Greater levels of fear of pain, pain-related anxiety, and fear-avoidance beliefs were significantly associated with greater pain intensity and disability [38]. In addition, higher levels of overly negative thoughts in response to pain or pain-related cues were associated with more pain intensity and disability levels [39]. The authors comment that an important observation in their reviews was that despite the very large number of studies that have been performed to evaluate the associations between pain-related factors and both pain and disability, the quality of the studies tended to be very low. These included issues in statistical analyses and reporting. These shortcomings made it difficult to carry out meta-analyses.

### 3.3. Other Sub-Fields

The paper by Yong Jei Lee and collaborators [40] is an example from criminology. The aim of their work was to show how many standard deviations in the number of crimes will change per a standard-deviation increase (or decrease) in the police-force size variable in the USA. They pooled standardized regression coefficients from 62 studies to estimate the overall effect size. The estimated pooled effect size was −0.030 (95% CI = −0.078 to 0.019). The nonsignificant and tiny mean effect size between police-force size and crime suggests that simply increasing police-force size may not help reduce crime, and if it does, then it does not reduce crime by much.

Meta-regression can be used in a meta-analysis to assess the relationship between study-level covariates and effect size [1]. Sanghee Park [41] applied meta-regression to study the effect of various study characteristics on the observed association between gender representation in the workforce and public-organization performance using the pooled $\beta$ as an effect-size index in 72 studies published between 1999 and 2017. Several covariates explained the variations in the reported $\beta$s. Unfortunately, the message of Park's article is hampered by an inadequate linkage between meta-regression theory and the reporting of the applied field of meta-analysis.

Yahui Tian and Jijun Yao [18] applied meta-analysis to analyze a total of 20 effect sizes from 11 articles on the impact of Chinese school resource investment on student performance. They found that the overall impact of school resources on student performance is significant (pooled $\beta$ = 0.093, 95% CI = 0.039 to 0.147). Since the standard regression coefficient was used as the effect size in this study, an increase of one standard deviation in school resource investment will increase student performance by 0.093 standard points. It should be noted that combining the effects of human, material and financial resources to an overall amount of resource investment in each study required multiple computational steps.

Standardized regression coefficients have also been applied in economics research. A paper published by Araujo et al. in 2020 provides a comprehensive synthesis of the evidence on macroprudential policies [42]. Drawing from 58 empirical studies, authors summarized the effects of macroprudential policy on several outcomes (e.g., credit, household credit, and house prices). The economic literature does not have a standard definition of the variables used to measure the effects of macroprudential policy. Enhancing the comparability of the effects across studies required the standardization approach to the regression coefficient between the macroprudential-policy variable and the corresponding outcome variable. The paper then used a meta-analysis framework to quantitatively synthesize estimated $\beta$s. In addition, meta-regression was used to examine how the $\beta$s varied with the study characteristics. Relying on $\beta$ as an effect size in meta-analysis techniques,

this paper demonstrated that on average, macroprudential-policy tools have statistically significant effects on credit.

## 4. Meta-Analysis Example

### 4.1. Research Question

With the rise in childhood obesity to epidemic portions across the world in the past few decades, many studies have sought to find out the long-term effects of childhood obesity on adulthood diseases [20]. The carotid artery intima-media thickness measured by ultrasound imaging represents a marker of preclinical atherosclerosis [43]. It correlates with vascular risk factors, associates with the severity of coronary artery disease, and predicts the likelihood of cardiovascular events in population groups. Several longitudinal cohort studies have tried to assess the relationship between childhood BMI or obesity and adulthood cIMT. The studies have shown conflicting results with some showing a positive association [43–47] while the other showing no significant association [48,49]. A systematic review [50], a pooled data analysis [51], and a meta-analysis [30] have also been conducted to assess these associations and have found qualitative positive associations between the two. Two of these studies [50,51] did not quantify the relationship, and the study of Ajala et al. [30] includes errors in pooling different effect-size metrics. With my example, I aim to clarify and quantify the association between childhood obesity and adulthood cIMT by combining evidence from the available studies.

### 4.2. Material and Methods

#### 4.2.1. Search Strategy

A literature search was carried out using Medline and Scopus from the year of inception to April 2022 with no language restrictions. The search strategy used a combination of medical subject headings and keywords to identify publications. The following search terms were used for the childhood-exposure variable: body mass index; BMI; child*; adolescen*; pediatric*; paediatric*. I combined these search terms with the search terms for the outcome variable: carotid intima-media thickness; intima-media thickness; carotid atherosclero*; carotid intima media; intimal-medial thickness; subclinical atherosclero*. Additional search terms were added to the aforementioned terms: prospective; retrospective; longitudinal; cohort; lifetime; long term; follow-up.

#### 4.2.2. Screening of Studies

The following criteria were used for the inclusion or exclusion of studies:

(a)   Type of study: prospective/retrospective longitudinal
(b)   Exposure: body mass index (BMI)
(c)   Age at measurement of body mass index: 2–19 years (childhood: 2–9 years; adolescence: 10–19 years)
(d)   Outcome: carotid intima-media thickness measured in adult ($\geq$20 years)
(e)   Length of follow-up: at least 5 years
(f)   Mode of ascertainment of exposure and outcome: all measurements taken by health professionals or trained investigators or from medical records.

Interventional studies, review articles and studies with selective groups, e.g., preterm babies, low- or high-birth-weight infants, obese children, etc. were excluded. In addition, studies using the categorized outcome variable cIMT and reporting odds ratios (ORs) or relative risks (RRs) were not included in this meta-analysis.

#### 4.2.3. Data Synthesis and Analysis

The effect sizes extracted from the original studies included correlation coefficients, mean differences, and unstandardized and standardized regression coefficients measuring the relationship between childhood and adolescent BMI and adult cIMT. Results from both unadjusted and adjusted analyses were included if they were included in the original

studies. Since BMI was measured in childhood or adolescence, most studies used a BMI variable standardized by age and sex.

I performed a meta-analysis to estimate the pooled effect of childhood and adolescent BMI on adult cIMT. In this analysis, standardized regression coefficients were used as effect-size estimates because different measurement methods and metrics were used across the original studies. Some studies provided the association between childhood BMI and adolescent BMI with maximum cIMT, whereas the other studies reported the association with mean cIMT measurements. Furthermore, childhood and adolescent BMI measurements were age- and sex-standardized using different growth charts. In these articles, the standard deviation of BMI (SD(BMI)) was equal to 1. If original studies presented correlation coefficients or unstandardized regression coefficients, then these were transformed to standardized regression coefficients using the formulas presented in Section 5.

### 4.3. Results

A total of 17 articles analyzing individuals from 16 different longitudinal cohort studies met the inclusion criteria and were included in the systematic review.

Table 1 reports the main characteristics of the 17 longitudinal studies included in this systematic review and meta-analysis. Sample sizes varied between 112 and 2628. Outcome (cIMT) was most frequently measured in individuals who were aged between 20–50 years.

**Table 1.** Characteristics of studies included in the meta-analysis.

| Study and Year of Publication | Country of Study | BMI Measured | | Sample Size | Baseline Age (Years) | Final Age (Years) |
|---|---|---|---|---|---|---|
| | | Childhood | Adolescent | | | |
| Ceponiene 2015 [52] | Lithuania | | ✓ | 380 | 12–13 | 48–49 |
| Davis 2001 [44] | United States | | ✓ | 725 | 8–18 | 33–42 |
| Du 2018 [53] | United States | ✓ | | 1052 | 9.8 (3.2) [a] | 23–43 |
| Ferreira 2004 [54] | Netherlands | | ✓ | 159 | 13–16 | 36.5 (0.6) [a] |
| Freedman 2004 [55] | United States | ✓ | ✓ | 513 | 4–17 | 23–40 |
| Hao 2018 [56] | United States | | ✓ | 626 | 10–18 | 24 [b] |
| Hosseinpanah 2021 [57] | Iran | | ✓ | 1295 | 10.9 (4.0) | 29.8 (4.0) [a] |
| Huynh 2013 [58] | Australia | | ✓ | 2328 | 7–15 | 26–36 |
| Johnson 2014 [59] | United Kingdom | | ✓ | 1273 | 15 | 60–64 |
| Juonala 2006 [60] | Finland | ✓ | | 1081 | 3–9 | 24–30 |
| Khalil 2013 [46] | India | ✓ | ✓ | 600 | 2, 11 | 33–38 |
| Lee 2008 [61] | South Korea | | ✓ | 256 | 16 | 25 |
| Oren 2003 [47] | Netherlands | | ✓ | 750 | 12–16 | 27–30 |
| Raitakari 2003 [43] | Finland | | ✓ | 1170 | 12–18 | 33–39 |
| Terzis 2012 [49] | Greece | | ✓ | 106 | 12–17 | 40.5 (1.1) [a] |
| Wright 2001 [62] | United Kingdom | ✓ | ✓ | 412 | 9, 13 | 50 |
| Yan 2017 [63] | China | | ✓ | 1252 | 6–18 | 27–42 |

[a] Mean (SD), [b] median.

Table 2 shows the effect sizes and computations applied to obtain the standardized regression coefficient β with standard error SE(β) for each evaluated study. Only 3 articles from the 17 included studies directly reported *β*-value estimated by linear-regression modeling [45,50,58]. These were used as the effect sizes in the meta-analysis. The standard error of *β* or *b* was not reported in several of the reviewed articles. In two studies, the authors were contacted to obtain SE(β) for their study [54,62]. In other studies, SE(β) was obtained from a confidence interval, or from a reported *p*-value or *t*-value of Wald's test statistic.

**Table 2.** Reported effect sizes and computations applied to obtain the standardized regression coefficient $\beta$ with standard error SE($\beta$) for each evaluated study. The numbers in columns refer to sub-sections of Chapter 5 where detailed formulas are provided.

| | Reported Effect Size | | Obtaining $\beta$ and SE($\beta$) | Combining within a Study | Estimating SD | Other Computations |
|---|---|---|---|---|---|---|
| Ceponiene [52] | | $b$ | 5.3.3 | 5.4 | 5.6.4 | |
| Davis [44] | $r$ | | 5.2.2 | 5.4 | | |
| Du [53] | | $b$ | 5.2.3 | | 5.6.4 | 5.7.1 |
| Ferreira [54] | $\beta$ | | 5.7.2 | | | 5.7.2 |
| Freedman [55] | $r$ | | 5.2.2 | 5.5 | | |
| Hao [56] | | $b$ | 5.2.5 | | | 5.7.3 |
| Hosseinpanah [57] | | $b$ | 5.2.5 | | 5.6.4 | |
| Huynh [58] | | $b$ | 5.3.3 | | 5.6.4 | |
| Johnson [59] | | $b$ | 5.3.3 | 5.4 and 5.5 | 5.6.2 | 5.7.1 |
| Juonala [60] | $r$ | | 5.2.2 | 5.4 | | |
| Khalil [46] | | $b$ | 5.3.3 | | 5.6.4 | |
| Lee [61] | | $b$ | 5.3.2 | 5.4 | 5.6.4 | 5.7.3 |
| Oren [47] | | $b$ | 5.3.3 | | 5.6.4 | |
| Raitakari [43] | | $b$ | 5.2.3 | | | |
| Terzis [49] | $\beta$ | | 5.3.2 | | | |
| Wright [62] | $\beta$ | | 5.7.2 | 5.4 | | 5.7.2 |
| Yan [63] | $r$ | | 5.2.2 | 5.4 | | |

$\beta$ = standardized regression coefficient, $r$ = correlation coefficient, $b$ = unstandardized regression coefficient, SD = standard deviation of cIMT or BMI.

A total of six studies reported effect sizes separately for males and females and one for age groups. Two studies analyzed data with repeated measurements where BMI was measured more than once at different age phases on the same children. For these studies I calculated composite effect sizes.

Standard deviations of BMI and cIMT were needed for the calculations of $\beta$ and SE($\beta$). In most of the studies SD(BMI) was 1. SD(cIMT) was not available in several of the evaluated articles. I applied the available data in eight articles to obtain the required standard deviation.

Tables 3 and 4 show the estimated $\beta$ effect sizes from the cohorts included in the meta-analysis. The first analysis included the studies in which the age at the assessment of BMI of the individuals was in childhood (Table 3, Figure 1, Childhood BMI). A 1 SD increase in childhood BMI leads to an increase of 0.047 SD (95% CI = 0.019 to 0.074; $p = 0.001$) in adult cIMT. Although statistically significant, this effect can be considered very small or negligible. The pooled standard deviation of cIMT among all the individuals included in this meta-analysis was 0.103. Using the Formula (8) of relationship between coefficients $b$ and $\beta$ from Section 5.7.1, a 1 SD increase in childhood BMI leads to an increase in adult cIMT by 0.047 × 0.103 = 0.005 mm (95% CI = 0.002 to 0.008 mm).

The second meta-analysis included studies where the individuals were in their adolescence at the time of the assessment of BMI (Table 4, Figure 1, Adolescent BMI). A 1 SD increase in adolescent BMI leads to a 0.127 SD (95% CI = 0.080 to 0.175; $p < 0.001$) or 0.127 × 0.103 = 0.013 mm (95% CI = 0.008 to 0.018 mm) increase in adult cIMT. According to this effect size, the relationship between the adolescent BMI and adult cIMT was small.

**Table 3.** Observed standardized regression coefficient $\beta$ with standard error SE($\beta$) and 95% confidence interval, sample size and weight in pooled analysis from seven studies estimating the relationship between childhood BMI and adult cIMT.

| | $\beta$ | SE($\beta$) | Lower Limit of 95% CI | Upper Limit of 95% CI | Sample Size | Weight (%) |
|---|---|---|---|---|---|---|
| Du 2018 | 0.054 | 0.024 | 0.007 | 0.101 | 1052 | 34.5 |
| Freedman 2004 | 0.100 | 0.063 | −0.023 | 0.223 | 246 | 5.0 |
| Johnson 2014 | 0.029 | 0.031 | −0.032 | 0.090 | 1273 | 20.7 |
| Juonala 2006 | 0.056 | 0.030 | −0.003 | 0.115 | 1078 | 22.1 |
| Khalil 2013 | 0.047 | 0.040 | −0.031 | 0.125 | 600 | 12.4 |
| Wright 2001 | −0.018 | 0.061 | −0.138 | 0.102 | 274 | 5.3 |
| Combined effect | 0.047 | 0.014 | 0.019 | 0.074 | 4523 | |

**Table 4.** Observed standardized regression coefficient $\beta$ with standard error *SE($\beta$)* and 95% confidence interval, sample size and weight in pooled analysis from 15 studies estimating the relationship between adolescent BMI and adult cIMT.

| | $\beta$ | *SE($\beta$)* | Lower Limit of 95% CI | Upper Limit of 95% CI | Sample Size | Weight (%) |
|---|---|---|---|---|---|---|
| Ceponie 2015 | 0.085 | 0.046 | −0.005 | 0.175 | 380 | 6.5 |
| Davis 2001 | 0.138 | 0.036 | 0.067 | 0.209 | 725 | 7.2 |
| Ferreira 2004 | 0.194 | 0.082 | 0.033 | 0.355 | 161 | 4.3 |
| Freedman 2004 | 0.184 | 0.048 | 0.090 | 0.278 | 825 | 6.4 |
| Hao 2018 | 0.243 | 0.025 | 0.194 | 0.292 | 496 | 7.8 |
| Hosseinpanah 2021 | 0.184 | 0.036 | 0.113 | 0.255 | 1295 | 7.2 |
| Huynh 2013 | 0.052 | 0.022 | 0.021 | 0.103 | 2328 | 8.0 |
| Johnson 2014 | 0.033 | 0.033 | −0.032 | 0.098 | 1273 | 7.3 |
| Khalil 2013 | 0.047 | 0.040 | −0.031 | 0.125 | 600 | 6.9 |
| Lee 2006 | 0.189 | 0.059 | 0.073 | .0305 | 256 | 5.7 |
| Oren 2003 | 0.046 | 0.010 | 0.026 | 0.066 | 750 | 8.4 |
| Raitakari 2003 | 0.090 | 0.030 | 0.031 | 0.149 | 1170 | 7.5 |
| Terzis 2012 | 0.098 | 0.095 | −0.088 | 0.284 | 106 | 3.7 |
| Wright 2001 | 0.062 | 0.064 | −0.063 | 0.187 | 242 | 5.4 |
| Yan 2017 | 0.266 | 0.026 | 0.215 | 0.317 | 1252 | 7.7 |
| Combined effect | 0.127 | 0.024 | 0.080 | 0.175 | 11859 | |

## 5. Detailed Description of Computations and Conversions

### 5.1. General

Often, evaluators confront the problem of different statistical methods and strategies being used to analyze the relationship between the response and explanatory variables [1,5,7]. The studies address the same broad question, and the reviewers want to include them in a meta-analysis. They need to convert the reported findings to a common index before they can proceed. The results expressed as linear-regression coefficients, correlation coefficients or mean differences can be re-expressed as standardized regression coefficients. This chapter provides formulas and procedures for computing standardized regression coefficients with standard errors from a variety of reported statistical data.

Studies vary in the usage of statistics to summarize the basic characteristics, sometimes using medians rather than means and sometimes using standard errors, confidence intervals, interquartile ranges and ranges to report variation. They also vary in the reporting of linear-regression models, sometimes reporting unstandardized or standardized regression coefficients, standard errors, or confidence intervals for coefficients, sometimes only *p*-values or models estimated in sub-groups. Inadequate data presentation and reporting problems are common in scientific articles in the evaluated articles [5,64].

In the literature review of the published meta-analysis using $\beta$ as the effect size (Section 3) and in my meta-analysis example (Section 4), I noticed that authors often confuse the unstandardized $b$ and the standardized $\beta$ coefficients in the description of their methods and in reported regression analysis tables. In addition, different symbols are used for these statistics in textbooks and statistical software. In healthcare and medicine, you can recognize a reporting error only if you are familiar both with the statistical methods used and the field under study. Interpreting the clinical meaning of the finding should reveal possible errors. For example, something is wrong if the article reports a standardized regression coefficient of 4.187.

To perform a meta-analysis of continuous data using $\beta$ as an effect-size index, researchers seek values of $\beta$ and SE($\beta$) from these numbers. Software procedures for performing meta-analyses using generic inverse-variance weighted averages take input data in the form of these estimates from each study [1,27].

When $\beta$ and SE($\beta$) are not directly available from the included article, procedures to estimate them from other reported data can be used. These calculations and conversions often require the standard deviation (SD) for response and explanatory variables. In several evaluated articles these are not given. In those articles they can be approximated using various methods depending on the data available in the article. In the following sections I describe how to calculate the standardized regression coefficient effect-size measure $\beta$ and its standard error SE($\beta$) in different research approaches and reporting styles of the original studies.

*5.2. Obtaining Standardized Regression Coefficients*

5.2.1. Coefficient $\beta$ Reported from a Linear-Regression Model

In an included article, when the standardized regression coefficient $\beta$ for the explanatory variable is reported from the estimated linear-regression model, it can be used directly as an effect size. An unadjusted or adjusted $\beta$ can be selected depending on the purpose of the meta-analysis. If the standard error of the $\beta$ is not reported, then it should be calculated from the other available information; see Section 5.3. Often models are estimated in subgroups, e.g., males and females separately. In these cases, effect sizes should be combined; see Section 5.4.

5.2.2. Correlation Coefficient r Reported

A study may only report a regression coefficient between the outcome and explanatory variables. In such a situation, the standardized regression coefficient is equal to the Pearson correlation coefficient $r$ between the variables. If the Spearman correlation coefficient is reported, then it can used as an approximation of $r$ and $\beta$. Basically, a Spearman coefficient is a Pearson correlation coefficient calculated with the ranks of the values of each of the two variables instead of their actual values. SE($\beta$) can be obtained using the formula

$$\text{SE}(\beta) = \frac{1 - r^2}{\sqrt{n - 1}} \tag{1}$$

where $r$ is the reported correlation coefficient and $n$ is the sample size [1,7].

5.2.3. Unstandardized Regression Coefficient $b$ Reported

If a study has estimated a simple linear regression $Y = a + b\,X$ or multivariable linear-regression model to report the regression coefficient $b$ between response $Y$ and explanatory variable $X$, then the $\beta$-value can be obtained by applying the formula

$$\beta = \frac{\text{SD}(X)}{\text{SD}(Y)} b \tag{2}$$

where SD($Y$) is the standard deviation of response variable and SD($X$) is the standard deviation of the exposure measure used in the study [5,23]. When SD($X$) or SD($Y$) are not

provided, the methods described in Section 5.6 can be used to calculate these statistics from other available data.

The standard error for $\beta$ is obtained as follows:

$$\mathrm{SE}(\beta) = \frac{\mathrm{SD}(X)}{\mathrm{SD}(Y)}\mathrm{SE}(b). \tag{3}$$

### 5.2.4. Mean Values of Outcome Variable Reported between Two Exposure Groups

When mean values of the response variable are compared between two groups (low- and high-exposure groups), the following statistics are usually given:

$n_1$ = sample size in group 1 and $n_2$ = sample size in group 2,
$M_1$ = mean value of response $Y$ in group 1 and $M_2$ = mean value in group 2
$\mathrm{SD}_1$ = standard deviation of $Y$ in group 1 and $\mathrm{SD}_2$ standard deviation in group 2
$\mathrm{SD}(Y)$ = full sample standard deviation oy outcome variable $Y$.

Now

$$b = M_1 - M_2$$

and the standard deviation of the dichotomous variable $X$ is

$$\mathrm{SD}(X) = \sqrt{\frac{n_1\,n_2}{(n_1 + n_2)^2}}$$

Using Equation (2) the standardized regression coefficient $\beta$ can be obtained by applying the formula

$$\beta = \sqrt{\frac{n_1\,n_2}{(n_1 + n_2)^2}}\;\frac{b}{\mathrm{SD}(Y)} = \sqrt{\frac{n_1\,n_2}{(n_1 + n_2)^2}}\;\frac{(M_1 - M_2)}{\mathrm{SD}(Y)}\,.$$

When SD(Y) is not reported in the article, it can be calculated using the formula

$$\mathrm{SD}(Y) = \sqrt{\frac{(n_1 - 1)\mathrm{SD}_1^2 + (n_2 - 1)\mathrm{SD}_2^2 + \frac{n_1\,n_2}{n_1 + n_2}\,(M_1 - M_2)^2}{n_1 + n_2 - 1}}\,.$$

From Formula (3) the standard error of $\beta$ is given by

$$\mathrm{SE}(\beta) = \frac{\mathrm{SD}(X)}{\mathrm{SD}(Y)}\mathrm{SE}(b) = \sqrt{\frac{n_1\,n_2}{(n_1 + n_2)^2}}\;\frac{\mathrm{SE}(b)}{\mathrm{SD}(Y)}\,.$$

SE($b$) can be obtained from the confidence interval of $b$ (=$M_1 - M_2$), or from the $t$-value or $p$-value of the $t$-test statistic to test the hypothesis $M_1 - M_2 = 0$. If these are not given, then the SE($b$) can be estimated by

$$\mathrm{SE}(b) = \mathrm{SE}(M_1 - M_2) = \mathrm{SD}_{pooled}(Y)\,\sqrt{\frac{1}{n_1} + \frac{1}{n_2}},$$

where the statistic

$$\mathrm{SD}_{pooled}(Y) = \sqrt{\frac{(n_1 - 1)\mathrm{SD}_1^2 + (n_2 - 1)\mathrm{SD}_2^2}{n_1 + n_2 - 2}} \tag{4}$$

is usually known as the pooled standard deviation of the outcome variable $Y$.

An alternative method is to covert the mean difference effect-size statistic

$$d = (M_1 - M_2)/\mathrm{SD}_{pooled}(Y)$$

to the regression coefficient $r$ [1,5,7]. This approach is not derived from the general relationship between $b$ and $\beta$ as described by Formula (2).

### 5.2.5. Mean Values of Outcome Variable Reported between More Than Two Exposure Groups

In some articles, authors have categorized using cut-off values of the explanatory variable $X$ to more than two groups with different ordered exposure levels, e.g., low, medium, high levels. Researchers have reported the mean response values by these groups and used an analysis of variance to compare the statistical significance of the mean values between these groups. A similar approach is to use dummy variables (indicator variables) in multivariable linear-regression modeling to indicate the groups of categorized explanatory levels and report the mean differences between these groups.

In the case of a categorized explanatory variable, a simple linear-regression line can be used to estimate the $\beta$ coefficient. In this approach, the group means of the outcome variable $Y$ are set as the dependent variable, and the selected values (contrasts) denoting the levels of the explanatory variable $X$ are the explanatory variable in the regression line [65]. Usually, values of 0, 1, 2, 3, ... $k$ are selected as contrast values when the explanatory variable is categorized to k ordered groups. The $\beta$ coefficient with SE($\beta$) can be obtained as follows:

$$\beta = \frac{\mathrm{SD}(contrast)}{\mathrm{SD}(Y)} \, b_c$$

and

$$\mathrm{SE}(\beta) = \frac{\mathrm{SD}(contrast)}{\mathrm{SD}(Y)} \, \mathrm{SE}(b_c),$$

where SD($contrast$) is the standard deviation of the selected contrast values, SD($Y$) is standard deviation of the outcome variable $Y$, and $b_c$ is the regression slope (regression coefficient of contrast values). This approach is also known as the linear trend test, and $b_c$ can be interpreted as the effect size for the trend between the exposure categories.

### 5.3. Obtaining Standard Error of Regression Coefficient from t-Value, p-Value or Confidence Interval

The standard error of the unstandardized ($b$) or standardized ($\beta$) regression coefficient can be obtained from a model output (if reported), from a reported confidence interval, from a $t$-statistic or a $p$-value to test the statistical significance of the coefficient or contacting authors of the original article. In addition, in an unadjusted analysis, SE($\beta$) can be obtained by applying Formula (1) from Section 5.2.2 for the standard error of the Pearson correlation coefficient $r$. I describe first how a $t$-statistic can be obtained from a $p$-value, then how SE can be obtained from a $t$-statistic, and finally how a confidence interval can be used to calculate SE. Meta-analysts may select the appropriate steps in this process according to what results are available to them.

#### 5.3.1. Standard Error from t-Value

The $t$-statistic tests the hypotheses that a regression coefficient ($b$ or $\beta$) equals to 0. The $t$-value is the ratio of the estimated regression coefficient to the standard error of the coefficient, i.e., $t = b/\mathrm{SE}(b)$ or $t = \beta/\mathrm{SE}(\beta)$. Thus, the standard error of the regression coefficient $b$ and $\beta$ can be obtained by applying the formulas

$$\mathrm{SE}(b) = \left| \frac{b}{t} \right|$$

and

$$\mathrm{SE}(\beta) = \left| \frac{\beta}{t} \right|$$

where $t$ is the observed value of the $t$-test to test the null hypothesis $H_0$: $b = 0$ (or $\beta = 0$).

### 5.3.2. Standard Error from *p*-Value

When only actual *p*-values obtained from *t*-tests are quoted, the corresponding *t*-value may be obtained from the *t*-distribution with $n - k - 1$ degrees of freedom, where $n$ is the sample size and $k$ is the number of explanatory variables in the regression model [65]. Standard statistical programs include a function to calculate the corresponding *t*-value. Difficulties are encountered when levels of significance are reported (such as $p < 0.05$ or even NS ('not significant', which usually implies $p > 0.05$) rather than actual *p*-values. A conservative approach would be to take the *p*-value at the upper limit (e.g., for $p < 0.05$ take $p = 0.05$, for $p < 0.01$ take $p = 0.01$, and for $p < 0.001$ take $p = 0.001$). However, this is not a solution for results that are reported as $p = $ NS, or $p > 0.05$.

### 5.3.3. Standard Error from Confidence Interval

If a 95% confidence interval is available for *b* or $\beta$, then the standard error SE can be calculated as:

$$\text{SE} = (upper\ limit - lower\ limit)/3.92 .$$

where *upper limit* and *lower limit* refer to the 95% confidence interval of the regression coefficient. For 90% confidence intervals 3.92 should be replaced by 3.29, and for 99% confidence intervals it should be replaced by 5.15.

### *5.4. Pooling Betas from Two or More Independent Sub-Groups*

In this section I consider cases where studies report data for two sub-groups of participants. For example, a study might report effect sizes separately for males and females. The defining feature here is that the sub-groups are independent of each other, so that each provides unique information. For this reason, it is possible to treat each sub-group as though it were a separate study [1]. This is one option to include the reported data into the meta-analysis. A second option is to compute a composite effect size for each study and use this in the meta-analysis. I consider this option in the following.

Let:

$\beta_1$ = standardized regression coefficient among females,
$\beta_2$ = standardized regression coefficient among males,
$\text{SE}(\beta_1)$ = SE of $\beta_1$,
$\text{SE}(\beta_2)$ = SE of $\beta_2$,
$W_1 = 1/(\text{SE}(\beta_1))^2$ weight for females,
$W_2 = 1/(\text{SE}(\beta_2))^2$ weight for males.

The combined effect of $\beta_P$ and $\text{SE}(\beta_P)$ can be obtained as follows:

$$\beta_P = (W_1\ \beta_1 + W_2\ \beta_2)/(W_1 + W_2)$$

$$\text{SE}(\beta_p) = \sqrt{\frac{1}{W_1 + W_2}} .$$

If the number of sub-groups is more than two, then the above formulas can be extended to the situation of several independent groups [1].

### *5.5. Pooling Effect Sizes Measured in More Than One Time Point*

Some studies may report findings where the outcome variable or the explanatory variable was measured more than once at different time points on the same participants. For example, in assessing the effect of BMI on cIMT, in one article BMI was measured at ages 3 and 9 years for the same children, and the effect on cIMT was reported separately for BMI at age 3 years and BMI at age 9 years. In another article, BMI was measured only at age 9 years, but cIMT was measured during adulthood twice, at ages 30 and 50 years. The effect size $\beta$ was reported separately for each adult age, but both measures were based on the same members of the cohort. This study design is also known as repeated measurements.

The effect sizes are not measured at independent groups but come from the same group of children or adolescents. Measurements at different time points are positively correlated. If the non-independent information is ignored in the combining of $\beta$s and their standard errors, then this will underestimate the standard error of the summary effect [66]. The procedure proposed by Bornstein et al. [1] can be used to combine the estimated $\beta$s across age phases (time points). This approach allows one to address the problem of repeated measurements, since the formula for the SE of combined effect sizes will take into account the correlation among the repeated measurements.

Let $\beta_j$ refer to the standardized regression coefficient estimated at the $j$ time point (age phase), $j = 1, 2, \ldots, m$. Thus, $m$ represents the number of different time points. Let the variance of coefficient $\beta_j$ be $V_j = (SE(\beta_j))^2$. The composite effect size $\beta_{ct}$ and the variance $V(\beta_{ct}) = (SE(\beta_{ct}))^2$ can be computed as

$$\beta_{ct} = \frac{1}{m}\left(\sum_{j=1}^{m}\beta_j\right)$$

and

$$V(\beta_{ct}) = \left(\frac{1}{m}\right)^2\left(\sum_{j=1}^{m}V_j + \sum_{j\neq k}\left(r_{jk}\sqrt{V_j}\sqrt{V_k}\right)\right), \tag{5}$$

where $r_{jk}$ is the correlation between effect sizes $\beta_j$ and $\beta_k$. Thus, the standard error of $b_{ct}$ is

$$SE(\beta_{ct}) = \sqrt{V(\beta_{ct})}.$$

If the variances $V_j$ are all equal to $V$ and the correlations are all equal to $r$, then Formula (5) of $V(\beta_{ct})$ simplifies to

$$V(\beta_{ct}) = \frac{1}{m}V(1 + (m-1)r).$$

The composite effect size of two correlated effect sizes ($m = 2$) is

$$\beta_{c2} = \frac{1}{2}(\beta_1 + \beta_2)$$

and variance

$$V(\beta_{c2}) = \frac{1}{4}\left(V_1 + V_2 + 2r\sqrt{V_1}\sqrt{V_2}\right).$$

*5.6. Estimating SD of Reponse and Explonatory Variables*

To calculate $\beta$ and $SE(\beta)$ from $b$ and $SE(b)$ using Formulas (2) and (3), the full-sample SD for $Y$ and $X$ are needed. Sometimes they are not available from the evaluated article. However, for the standard deviations there is an approximate or direct algebraic relationship with other measures of variation, so that it is possible to obtain the required statistic even if it is not published in the article.

5.6.1. SD from Ranges

If *minimum* and *maximum* values of response variable are given, then they can be used to estimate standard deviation. Ranges (*maximum–minimum*) are very unstable and, unlike other measures of variation, increase when the sample size increases. They describe the extremes of observed outcomes rather than the average variation. One common approach has been to make use of the fact that, with normally distributed data, 95% of values will lie within 2 SD of either side of the mean. The standard deviation SD may therefore be estimated to be approximately one-quarter of the typical range of data values, i.e., (*maximum- minimum*)/4. This method may not work well in practice when the sample size

is large ($n > 70$) [67,68]. To overcome this problem, the following improved range rule of thumb is often used

$$SD(Y) = \frac{(maximum - minimum)}{6}.$$

Alternative methods have been proposed to estimate SDs from ranges [67–69].

### 5.6.2. SD from Interquartile Range

An interquartile range is defined as the difference between the *upper quartile* and *lower quartile* (75th and 25th percentiles) of the analyzed variable. It describes where the central 50% of participants' outcomes lie. When sample sizes are large and the distribution of the outcome is similar to the normal distribution, the width of the interquartile range will be approximately 1.35 SD. Thus

$$SD(Y) = \frac{(upper\ quartile - lower\ quartile)}{1.349}.$$

Wan and colleagues [68] provided a sample-size-dependent extension to the formula for approximating the SD using the interquartile range.

### 5.6.3. SD from SE

If the standard error $SE(Y)$ of the response variable $Y$ is reported, then $SD(Y)$ is given by

$$SD(Y) = \sqrt{n}SE(Y), \tag{6}$$

where $n$ is the sample size.

If $SE(Y)$ is not given, then it can be estimated from the confidence interval for the mean value of response $Y$. Then by (6)

$$SE(Y) = \frac{\sqrt{n}\ (upper\ limit - lower\ limit)}{3.92}$$

where *upper limit* and *lower limit* refer to the 95% confidence interval for the mean value of $Y$.

### 5.6.4. Pooling Groups to Obtain SD

Sometimes it is necessary to combine two reported sub-groups into a single group to obtain the full-sample SD of $Y$ or $X$. For example, a study may report results separately for men and women. The following formula can be used to combine standard deviations into a full sample SD:

$$SD_{full\ sample} = \sqrt{\frac{(n_1 - 1)SD_1^2 + (n_2 - 1)SD_2^2 + \frac{n_1\ n_2}{n_1 + n_2}\ (M_1 - M_2)^2}{n_1 + n_2 - 1}} \tag{7}$$

where $n_1$ and $n_2$ are sample sizes, $SD_1$ and $SD_2$ are standard deviations, and $M_1$ and $M_2$ are mean values of groups 1 and 2.

Note that the rather complex-looking Formula (7) for the SD produces the SD of outcome measurements as if the combined group had never been divided into two. This SD is different from the usual pooled SD in (4) that is used to compute a confidence interval for a mean difference or as the denominator in computing the standardized mean difference. The pooled SD provides a within-sub-group SD rather than an SD for the combined group, and thus provides an underestimate of the desired SD.

When there are more than two groups to combine, the simplest strategy is to apply the above Formula (7) sequentially (i.e., combine Group 1 and Group 2 to create Group '1 + 2', then combine Group '1 + 2' and Group 3 to create Group '1 + 2 + 3', and so on).

There are also other methods to estimate the full-sample standard deviation of a variable when findings are reported only in sub-groups [70]. However, these more complex calculation formulas need additional information that is not necessarily available.

*5.7. Other Topics*

5.7.1. Interpretation with the Unit of Measurement of the Outcome Variable

The results represented by the standardized coefficient $\beta$ can also be expressed in terms of the original measurement unit of the outcome variable. By Equation (2) the standardized regression coefficient $\beta$ represents how many standard deviation units the outcome variable $Y$ will change per a standard deviation increase in the explanatory variable $X$. If SD($X$) equals to one, then Equation (2) gives

$$b = \frac{SD(Y)}{SD(X)}\beta = SD(Y)\beta \tag{8}$$

If we know the standard deviation of the outcome variable $Y$, then we can estimate how much variable $Y$ will change per one standard deviation change in the explanatory variable. For example, in the previous chapter, the outcome variable was cIMT and the predictor factor was childhood or adolescent BMI. A positive $\beta$-value equal to 0.08 demonstrates that a one-standard-deviation increase in childhood BMI leads to a cIMT increase (in mm) equal to the standard deviation SD(cIMT) of cIMT multiplied by 0.08. Further, if the pooled SD of cIMT from all the included studies is 0.10 mm, then we obtain a cIMT increase (in mm) of $\beta$ SD(cIMT) = 0.08 × 0.10 mm = 0.008 mm per one-standard-deviation increase in childhood BMI.

5.7.2. Log-Transformed Data

The standardized regression coefficient can be used as an effect-size index to pool both raw and log-transformed outcomes (or explanatory variables). The standardized regression coefficient does not estimate effects on the original scales of variables but refers to the standard deviations of the variables.

When an original study involves an outcome variable with a skewed distribution, the reported data can sometimes be a mixture of results presented on the raw scale and results presented on the logarithmic scale [71]. A common approach to dealing with skewed outcome data is to take a logarithmic transformation of each observation and to conduct the regression modeling using log-transformed values. However, for ease of interpretation, basic characteristics are often reported using the initial unit of measurement (raw scale). When the estimated regression coefficient $b$ and SE($b$) are estimated for the log-transformed variable ($lnY$), then we need SD($lnY$) to calculate the standardized regression coefficient (and SE) for the ($lnY$).

To obtain the approximate standard deviation of the outcome variable $Y$ on the log-transformed scale, the following formula can be used:

$$SD(lnY) = \sqrt{\ln(1 + \frac{SD_Y^2}{M_Y^2})}\,,$$

where $SD_Y$ is the reported standard deviation and $M_Y$ is the mean value of variable $Y$ [71].

5.7.3. Contacting Authors

Missing data and clarification about the statistics required for the meta-analysis could be sought from the authors of the original studies. Although challenging, obtaining additional data through author contacts can enable reviewers to synthesize data more readily and completely. Authors of more recent studies are more likely to be located and provide data compared to authors of older studies [72]. Contacting authors may be time-consuming, not only for meta-analysts but also for the study authors. They need to locate

the data before being able to provide the statistics. However, when needed, authors of studies with missing or discrepant data should be contacted.

### 5.7.4. Imputing Missing Statistics

Missing SEs of the effect size or standard deviations of the main variables are a common feature of meta-analyses of continuous outcome data. When none of the methods described in the previous sections allow the calculation of the SEs or SDs from the study report (and the information is not available from the authors), then a meta-analysist may be forced to impute ('fill in') the missing data if they are not to exclude the study from the meta-analysis [73,74].

There are several obvious advantages to imputing the missing data compared to a meta-analysis using only the studies with reported statistics. Imputing allows the inclusion of more studies, thus reducing the overall standard error of the estimate of the effect size, compared to using only studies reporting information [74]. The simplest imputation is to borrow the missing value of a statistic from one or more other studies. If several candidate SDs are available, reviewing researchers should decide whether to use their average, the highest, a 'reasonably high' value, or some other strategy. Choosing a higher SD down-weights a study and yields a wider confidence interval. Thus, choosing a higher SD will bias the result towards a lack of effect.

## 6. Discussion

Many original studies addressing the same research question are relatively small and differ in their statistical content for various reasons. It is important to have practicable research methods to pool findings from different studies to quantify the relationships between predictor variables and outcomes. This article summarizes the procedures of applying the standardized regression coefficient β for the synthesis of an association between a quantitative dependent variable and one focal explanatory factor when the measurement methods and controlling of other potential covariates varies between the reviewed studies. I described how it is possible to use β as a workable effect-size statistic that can be applied to the research findings of interest. I applied this method in a systematic review of studies to provide evidence for the relationship between childhood and adulthood BMI and cIMT in adult life using effect sizes that were continuous variables.

### 6.1. Issues Regarding the Conduction of Standardized Regression Coefficient

There are issues in combining and analyzing the standardized regression coefficients. These potential problems are related to the variation of the variables to be controlled, multiple conversions of effect sizes and data presentation in the original studies. Riley et al. [75] gives the following detailed list of challenges for the meta-analysis of multivariable findings:

(a) Different types of effect measures (e.g., correlation coefficients, regression coefficients, risk ratios, odd ratios and mean differences), which are not necessarily comparable.
(b) Estimates without standard errors, which is a problem because meta-analysis methods typically weight each study by their standard error.
(c) Estimates relating to various time points of the outcome occurrence or measurement.
(d) Different methods of measurement for explanatory variables and outcomes.
(e) Various sets of adjustment factors.
(f) Different approaches to handling continuous explanatory variables (e.g., categorization, linear, non-linear trends, log-transforms), including the choice of cut point value when dichotomizing continuous values into "high" and "normal" groups.

In addition, shortcomings in the reporting of the included publications makes meta-analysis challenging.

### 6.1.1. Different Adjusted Covariates

Several researchers have discussed the problem that the covariates in multiple regression models can vary across studies [9,10,76,77]. In the original studies, multivariable regression models have been used to estimate the independent effect of the main explanatory variable on a response variable when confounding factors are controlled. It is ideal, but highly unlikely, that the estimated effect sizes from different studies are adjusted for the same confounding factors (covariates) [75]. In the synthesis of standardized regression coefficients, pooling the independent effects of the focal explanatory variable is still important to obtain an estimate of the association between the explanatory and outcome variables. If adjustment factors are omitted, then the observed effects could be too optimistic. Estimates adjusted for a different set of covariates creates difficulty in interpreting meta-analysis results. To overcome this issue, Riley et al. [75] recommended considering meta-analysis only on those estimates that are adjusted for at least a predefined minimum core set of established covariates. This core set of covariates for the outcome can be defined in consultation with experts. In addition, separate meta-analyses could be performed for unadjusted and adjusted prognostic effect estimates. Even when control variables and other predictors differ between studies, the pooling of $\beta$s still provide useful information about the size of the effect. Generally, meta-analysis results will be most interpretable, and therefore useful, when a separate meta-analysis is undertaken for groups of "similar" prognostic variables. However, it is evident that enhancements to the associated synthesis methodologies are urgently needed. Becker and Wu described existing methods of analysis and presented a multivariate generalized least-squares approach to the synthesis of regression coefficients [9]. Yoneoka and Henmi [78] extended this approach and proposed a synthesis methodology for regression results under different covariate sets by using a generalized least-squares method that includes bias-correction terms. However, the combination will be exponentially complex as the number of covariates increases.

### 6.1.2. Several Transformations and Conversions

Converting reported effect sizes from estimated regression models to β coefficients requires several transformations, and on some occasions, data imputation or manipulation of the statistical information available in a report. Converting from a reported effect size to a β coefficient may not go smoothly. The data conversion begins with the extraction of the information from an original article. This can be frustrating if the article fails to report the statistics required in the formulas for computing the β-values. In addition, understanding the analysis approach and methods used in the original article may sometimes be difficult.

However, effect-size transformation provides an opportunity to make a study available for meta-analysis. Data transformation facilitates the compatibility between studies with same research question. The question of whether or not it is appropriate to combine effect sizes from studies that used different statistical methods or metrics must be considered on a case-by-case basis. It only makes sense to compute a summary effect from studies that we assess to be comparable in a meaningful way. If it would be comfortable to combine these studies if they had used the same method, then the fact that they used different methods or metrics should not be an obstacle [1]. Although not without concerns, this approach produces reasonably similar results from other methods [9]. The decision to use these conversions is often better than the alternative, which is to simply omit studies that happened to use an alternative measure effect size. This would involve the loss of information, and possibly result in a biased sample of studies.

### 6.1.3. Insufficient Reported Data

One obstacle in conducting a meta-analysis is insufficiently reported data in evaluated articles to compute effect-size estimates. Detailed descriptive statistics of the variables under study are not given in all articles, and standard errors for regression coefficients are not always available [5]. In some cases, the incomplete reporting of statistics in the studies limits or prevents the use of these studies in the systematic review [4].

The validity and practical utility of observational research critically depends on good study design, appropriate analysis methods and high-quality reporting and data presentation [79,80]. In reviewed studies, the reporting of observational findings often exhibits serious shortcomings [80]. An efficient way to help readers to extract the necessary data is to develop guidance documents of data presentation that are disseminated to the research community at large. We need a much more structured framework in scientific reporting, which emphasizes that today's scientific evidence is based on the synthesis of studies reporting findings with similar effect-size indices [64]. Especially, the reporting of estimated multivariable regression models needs attachments such as tables and figures reporting descriptive statistics about the distributions of the response variables and explanatory variables. This would help other researchers to utilize the results in their approaches to summarize and meta-analyze the magnitude of the effects.

In the future, this issue could be even more pronounced with the application of machine-learning methods. Machine-learning methods do not provide effect sizes (indices) that can be combined or that are interpretable for clinicians.

### 6.2. Meta-Analysis of Association between BMI and cIMT

In the illustrative meta-analysis study, I aimed to provide evidence for the relationship between childhood and adulthood BMI and cIMT in adult life using the β coefficient as the effect-size index. This approach helped us to quantify the relationship. Findings from my meta-analysis indicate that elevated childhood and adult BMI is associated with only a modest increase in carotid intima-media thickness in adult life. Adolescent BMI had a marginally stronger relation with adult cIMT than childhood BMI. In general, the results are consistent with those of previous systematic reviews [30,50,51]. However, these previous reviews used different approaches and different effect sizes that were not entirely suitable for the research problem.

The quantification of the associations made in my study showed that these significant effect sizes reflect only a modest increase in cIMT. These small increases in cIMT might be difficult to equate to true clinical significance. One explanation for these small increases in cIMT might be the younger age of most of the participants at the time of the evaluation of cIMT. Longer follow-ups might be necessary to demonstrate the utility of these findings.

There were some limitations in this study. First, BMI is not an ideal measure of obesity and caution must be used while interpreting these results. However, BMI remains to be the most basic and the most commonly used tool for the assessment of obesity due to the ease of measurement, the ease of interpretation of the results, and its low expense. While tests like dual-energy X-ray absorptiometry, computed tomography and magnetic resonance imaging are the gold standards for assessing regional obesity, their use is limited by their high cost, lack of availability, and non-portable, required equipment for use in routine clinical practice. In the presence of these constraints, BMI has remained a valuable tool for the assessment of obesity in clinics, hospitals, and in large epidemiological studies. Second, multiple transformations of effect sizes for some studies were needed to translate them into one common effect size. A comparison of the effect sizes produced by different statistical techniques is a challenge for readers and especially those wanting to carry out a meta-analysis [Nieminen 2013]. However, it is often informative to translate the effect-size results from the original studies to one effect-size index to reveal the pooled effects.

### 7. Conclusions

Statistical methods used in studies should not be the basis for the inclusion of the studies in a meta-analysis. The same research question can be analyzed with different statistical methods. Measurements methods, data transformations, descriptive statistics, and statistical inference methods may vary between studies. In addition, authors may focus on reporting in different ways. This reflects the reality when reviewing published research articles and trying to summarize the findings from observational studies. The proposed approach based on standardized regression coefficients provides a workable

effect-size index that can be applied to the systematic review of diverse multivariable studies with quantitative outcomes. I applied this method in a meta-analysis providing evidence that BMI in childhood and adult have a minimal effect on adult cIMT. As the observed effect sizes are very low, they are unlikely to correlate with clinically significant differences. In addition, from the public-health point of view, the small effect sizes suggest that the introduction of interventions to reduce obesity in childhood might not have a high impact on the subsequent cIMT measurements in young-adult life.

**Funding:** This research received no external funding.

**Institutional Review Board Statement:** Not applicable.

**Informed Consent Statement:** Not applicable.

**Data Availability Statement:** Data for the meta-analyses are presented in Tables 3 and 4.

**Acknowledgments:** I am grateful to Jasleen Kaur for her valuable and constructive suggestions during the development of this research work, especially in the literature search and screening of studies.

**Conflicts of Interest:** I declare no conflict of interest.

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
