# Peer review of "Application of Standardized Regression Coefficient in Meta-Analysis"

_biomedinformatics, doi:10.3390/biomedinformatics2030028_

Round 1

Reviewer 1 Report

Here is presented a review titled: Application of standardized regression coefficient in meta-analysis. The author has provided a comprehensive review of how variability in analysis of association between variables in different multivariate models lacks consistency in findings. The author presents case studies support of their arguments succinctly.

Author Response

Here is presented a review titled: Application of standardized regression coefficient in meta-analysis. The author has provided a comprehensive review of how variability in analysis of association between variables in different multivariate models lacks consistency in findings. The author presents case studies support of their arguments sufficiently.

Thank you.

Reviewer 2 Report

This review article by Pentti Nieminen describes and summarizes the application of standardized regression coefficient in meta-analysis in public health, psychology and various other fields and further demonstrate with an example of research syntheses using published articles on association between childhood obesity in terms of BMI and adult carotid intima-media thickness (cIMT). Finally, this review article offers a recommended guidelines on the use of standardized regression coefficient in pooling findings. This review guidelines are highly useful for scientists routinely performing meta-analysis and I feel that the manuscript is suitable for publication after the following issues are addressed:

Minor

  1. Line 10: “The main purpose of this paper is to illustrate the main procedures in summarizing” can be changed to restrict the use of word “main” and refer to review article instead of paper. Recommended change is “The main purpose of this review article is to illustrate the procedures in summarizing”.
  2. Line 18: The use of word “Synthesis” should be consistent, compared to line 14.
  3. Recommended: Keywords at Line 18, can also include “statistics”
  4. The use of CIMT at line 63, should be corrected to cIMT.
  5. The author is recommended to refer to Figure 1 as an example of “forest plot” an illustration for the paragraph from Line 133-139, this will enable readers the ease of understanding.
  6. Line: 330. The use of ‘We’ in a single author paper is debatable! Kindly acknowledge other co-authors contribution if any or refrain from using ‘We’. (Follow up:) Line 368: “met our inclusion criteria”; Line 390, 394, 396, 541, 586, 589, 850 and wherever applicable, please cross verify the usage of ‘we’ and ‘our’

Author Response

This review article by Pentti Nieminen describes and summarizes the application of standardized regression coefficient in meta-analysis in public health, psychology and various other fields and further demonstrate with an example of research syntheses using published articles on association between childhood obesity in terms of BMI and adult carotid intima-media thickness (cIMT). Finally, this review article offers a recommended guidelines on the use of standardized regression coefficient in pooling findings. This review guidelines are highly useful for scientists routinely performing meta-analysis and I feel that the manuscript is suitable for publication after the following issues are addressed:

Thank you.

Line 10: “The main purpose of this paper is to illustrate the main procedures in summarizing” can be changed to restrict the use of word “main” and refer to review article instead of paper. Recommended change is “The main purpose of this review article is to illustrate the procedures in summarizing”.

I appreciate this comment. I have now used the recommended formulation “The main purpose of this review article is to illustrate the procedures in summarizing …”.

Line 18: The use of word “Synthesis” should be consistent, compared to line 14.

I have replaced "syntheses" with "synthesis" in line 14.

Recommended: Keywords at Line 18, can also include “statistics”

I agree and I have added "statistics" to the Keywords.

The use of CIMT at line 63, should be corrected to cIMT.

I have corrected the typo.

The author is recommended to refer to Figure 1 as an example of “forest plot” an illustration for the paragraph from Line 133-139, this will enable readers the ease of understanding.

I agree and I am grateful to the reviewer for the helpful suggestion. I have now referred to Figure 1 as an example of forest plot. I note in the text that “Figure 1 gives two examples of forest plots”.

Line: 330. The use of ‘We’ in a single author paper is debatable! Kindly acknowledge other co-authors contribution if any or refrain from using ‘We’. (Follow up:) Line 368: “met our inclusion criteria”; Line 390, 394, 396, 541, 586, 589, 850 and wherever applicable, please cross verify the usage of ‘we’ and ‘our’.

I thank the reviewer for drawing my attention to the incorrect use of “we” or “our”. I have replaced these with “I” and “my”, or I have used a different wording.